# Distinct Phenotypes of Kidney Transplant Recipients in the United States with Limited Functional Status as Identified through Machine Learning Consensus Clustering

**DOI:** 10.3390/jpm12060859

**Published:** 2022-05-25

**Authors:** Charat Thongprayoon, Caroline C. Jadlowiec, Wisit Kaewput, Pradeep Vaitla, Shennen A. Mao, Michael A. Mao, Napat Leeaphorn, Fawad Qureshi, Pattharawin Pattharanitima, Fahad Qureshi, Prakrati C. Acharya, Pitchaphon Nissaisorakarn, Matthew Cooper, Wisit Cheungpasitporn

**Affiliations:** 1Division of Nephrology and Hypertension, Department of Medicine, Mayo Clinic, Rochester, MN 55905, USA; qureshi.fawad@mayo.edu; 2Division of Transplant Surgery, Mayo Clinic, Phoenix, AZ 85054, USA; jadlowiec.caroline@mayo.edu; 3Department of Military and Community Medicine, Phramongkutklao College of Medicine, Bangkok 10400, Thailand; 4Division of Nephrology, University of Mississippi Medical Center, Jackson, MS 39216, USA; pvaitla@umc.edu; 5Division of Transplant Surgery, Mayo Clinic, Jacksonville, FL 32224, USA; mao.shennen@mayo.edu; 6Division of Nephrology and Hypertension, Department of Medicine, Mayo Clinic, Jacksonville, FL 32224, USA; mao.michael@mayo.edu; 7Renal Transplant Program, University of Missouri-Kansas City School of Medicine/Saint Luke’s Health System, Kansas City, MO 64108, USA; napat.leeaphorn@gmail.com; 8Department of Internal Medicine, Faculty of Medicine, Thammasat University, Pathum Thani 12120, Thailand; 9School of Medicine, University of Missouri-Kansas City, Kansas City, MO 64108, USA; fmaqureshi@gmail.com; 10Division of Nephrology, Texas Tech Health Sciences Center El Paso, El Paso, TX 79905, USA; prakrati.c.acharya@gmail.com; 11Department of Medicine, Division of Nephrology, Massachusetts General Hospital, Harvard Medical School, Boston, MA 02114, USA; pitch.nissa@gmail.com; 12Medstar Georgetown Transplant Institute, Georgetown University School of Medicine, Washington, DC 21042, USA; matthew.cooper@gunet.georgetown.edu

**Keywords:** functional status, disability, disabled, kidney transplant, transplantation, clustering, machine learning

## Abstract

Background: There have been concerns regarding increased perioperative mortality, length of hospital stay, and rates of graft loss in kidney transplant recipients with functional limitations. The application of machine learning consensus clustering approach may provide a novel understanding of unique phenotypes of functionally limited kidney transplant recipients with distinct outcomes in order to identify strategies to improve outcomes. Methods: Consensus cluster analysis was performed based on recipient-, donor-, and transplant-related characteristics in 3205 functionally limited kidney transplant recipients (Karnofsky Performance Scale (KPS) < 40% at transplant) in the OPTN/UNOS database from 2010 to 2019. Each cluster’s key characteristics were identified using the standardized mean difference. Posttransplant outcomes, including death-censored graft failure, patient death, and acute allograft rejection were compared among the clusters Results: Consensus cluster analysis identified two distinct clusters that best represented the clinical characteristics of kidney transplant recipients with limited functional status prior to transplant. Cluster 1 patients were older in age and were more likely to receive deceased donor kidney transplant with a higher number of HLA mismatches. In contrast, cluster 2 patients were younger, had shorter dialysis duration, were more likely to be retransplants, and were more likely to receive living donor kidney transplants from HLA mismatched donors. As such, cluster 2 recipients had a higher PRA, less cold ischemia time, and lower proportion of machine-perfused kidneys. Despite having a low KPS, 5-year patient survival was 79.1 and 83.9% for clusters 1 and 2; 5-year death-censored graft survival was 86.9 and 91.9%. Cluster 1 had lower death-censored graft survival and patient survival but higher acute rejection, compared to cluster 2. Conclusion: Our study used an unsupervised machine learning approach to characterize kidney transplant recipients with limited functional status into two clinically distinct clusters with differing posttransplant outcomes.

## 1. Introduction

Functional status, or the capacity to perform daily activities to meet basic needs and maintain health and wellbeing, is recognized as a useful tool for evaluating patients in various clinical settings, including patients undergoing organ transplantation [1,2,3,4]. Among kidney transplant recipients, significant associations between low functional status and poor outcomes after kidney transplant have been reported, including increased mortality [5,6,7,8,9]. In the United States (U.S.), the Karnofsky Performance Scale (KPS), a physician-reported measure of functional status ranging from 0 to 100%, [10] is routinely collected among transplant candidates and recipients and has become a requirement by the Organ Procurement and Transplantation Network (OPTN) for risk adjustment of transplant outcomes [4,11,12,13,14].

Prior studies have demonstrated an association between low KPS scores and poor outcomes after kidney transplantation, including mortality and all-cause graft loss [4,11]. Additionally, recipients with low functional scores are often found to be ineligible for kidney transplantation due to a combination of limited transplant resources and anticipation of a lessened survival benefit post kidney transplantation. As a result, the majority of kidney transplant surgeries in the U.S. are performed for kidney transplant recipients with satisfactory functional status [11]. Only 3% of kidney transplant surgeries are performed for patients with KPS of ≤40% [11]. While patients with a low functional status have a higher risk of post-operative complications and death following kidney transplant as compared to those with higher physical functioning scores, this risk is likely still less than remaining on dialysis [1,11,15]. Furthermore, kidney transplant patients, including recipients with low functional status, are likely heterogenous, and there are many factors, including recipient, donor, and transplant-related variables, that can result in varying outcomes contrary to what has historically been reported in the literature [11,16,17].

Artificial intelligence and machine learning (ML) have been utilized to provide clinical decision support tools and individualize patient care, including in organ transplantation [18,19,20,21,22,23,24]. Unsupervised consensus clustering is ML applied to discover novel data patterns and distinct subtypes [25,26,27]. It can discover similarities and heterogeneities among various data variables and distinguish them into clinically meaningful clusters [25,26]. Recent studies have demonstrated that distinct subtypes identified by the ML consensus clustering approach can forecast different clinical outcomes [28,29]. Given data on characteristics of kidney transplant recipients with KPS of ≤40% in the U.S. are limited, the application of ML consensus clustering approach may provide a novel understanding of unique phenotypes of disabled kidney transplant recipients with distinct outcomes in order to identify strategies to improve their outcomes. 

In this study, we analyzed the United Network for Organ Sharing database (UNOS)/OPTN database from 2010 through 2019 using an unsupervised ML clustering approach to identify distinct clusters of kidney transplant recipients whose functional status at transplant were impaired (KPS of ≤40%) and assess clinical outcomes among these unique clusters. 

## 2. Methods

### 2.1. Data Source and Study Population

This study was conducted using the UNOS/OPTN database to screen adult kidney-only transplant recipients in the United States from 2010 to 2019 with low functional status. Low functional status recipients were defined as having a KPS of ≤40% at the time of kidney transplantation. This study received approval from the Mayo Clinic Institutional Review Board (IRB number 21-007698). 

### 2.2. Data Collection

The following recipient-, donor-, and transplant-related variables in the OPTN/UNOS database were abstracted for inclusion in cluster analysis: recipient age, sex, race, body mass index (BMI), kidney retransplant, dialysis duration, causes of end-stage kidney disease, comorbidities, panel reactive antibody (PRA), hepatitis C, hepatitis B and human immunodeficiency virus (HIV) serostatus, KPS, working income, insurance, US residency status, education, serum albumin, kidney donor type, ABO incompatibility, donor age, sex, race, history of hypertension in donor, kidney donor profile index (KDPI), HLA mismatch, cold ischemia time, kidney on pump, delay graft function, allocation type, Ebstein–Barr virus (EBV) and Cytomegalovirus (CMV) status, induction, and maintenance immunosuppression. 

Functional status at the time of transplant for kidney transplant recipients was defined by center-reported KPS. KPS, as shown in Appendix A, is a categorical classification system with progressive but arbitrary increase in assigned performance status at 10% intervals without use of the intervening numbers, so it was treated as ordinal for statistical purposes. Patients were further categorized into four groups: normal (80–100%), capable of self-care (70%), requires assistance (50–60%), and disabled (10–40%). Among those with KPS of ≤40%, detailed definitions as following: KPS of 40%: disabled and requires special care and assistance; KPS of 30%: severely disabled, hospital admission is indicated although death is not imminent; KPS of 20%: very sick, hospital admission and active supportive treatments are necessary; KPS of 10%: moribund, fatal processes progress rapidly. All extracted variables had missing data less than 5% (Appendix A). Missing data were imputed through multivariable imputation by chained equation (MICE) method [30]. 

### 2.3. Clustering Analysis

Unsupervised ML was applied by conducting a consensus clustering approach to categorize clinical phenotypes of functionally disabled kidney transplant recipients (KPS at transplant of ≤40%) [31]. A pre-specified subsampling parameter of 80% with 100 iterations and the number of potential clusters (k) ranging from 2 to 10 were used to avoid producing an excessive number of clusters that would not be clinically useful. The optimal number of clusters was determined by examining the consensus matrix (CM) heat map, cumulative distribution function (CDF), cluster-consensus plots with the within-cluster consensus scores, and the proportion of ambiguously clustered pairs (PAC). The within-cluster consensus score, ranging between 0 and 1, was defined as the average consensus value for all pairs of individuals belonging to the same cluster [32]. A value closer to one indicates better cluster stability. PAC, ranging between 0 and 1, was calculated as the proportion of all sample pairs with consensus values falling within the predetermined boundaries [33]. A value closer to zero indicates better cluster stability [33]. The PAC was calculated using two criteria (1) the strict criteria consisting of a predetermined boundary of (0, 1), where a pair of individuals who had a consensus value >0 or <1 was considered ambiguously clustered, and (2) the relaxed criteria consisting of a predetermined boundary of (0.1, 0.9), where a pair of individuals who had consensus value >0.1 or <0.9 was considered ambiguously clustered [33]. The detailed consensus cluster algorithms used in this study for reproducibility are provided in Appendix A. 

#### Outcomes

Posttransplant outcomes consisted of death-censored graft survival, patient survival within 1 and 5 years after kidney transplant, and acute allograft rejection within 1 year after kidney transplant. We defined death-censored graft failure as the need for dialysis or kidney retransplant, while censoring patients for death or at last follow-up date reported to the OPTN/UNOS database. 

### 2.4. Statistical Analysis 

After an individual functionally impaired kidney transplant patient was assigned a cluster using the consensus clustering approach, statistical analyses were subsequently performed to compare the characteristics and outcomes among the assigned clusters. The differences in clinical characteristics among the assigned clusters were tested using Student’s t-test for continuous variables and Chi-squared test for categorical variables. The key characteristics of each cluster were determined using the standardized mean difference between each cluster and the overall cohort with the cut-off of >0.3. 

The difference in posttransplant outcomes were evaluated among the assigned clusters. The hazard ratios (HR) for death-censored graft failure and patient death based on the assigned clusters were obtained using Cox proportional hazard analysis. Because the OPTN/UNOS database did not specify the date of allograft rejection occurrence, the odds ratio (OR) for 1-year allograft rejection based on the assigned clusters was obtained using logistic regression analysis. The hazard ratio or odds ratio were not adjusted for the between-cluster difference in clinical characteristics since an unsupervised consensus clustering approach was conducted to purposefully generate clinically distinct clusters. 

All analyses were carried out using R, version 4.0.3 (RStudio, Inc., Boston, MA, USA; http://www.rstudio.com/, accessed on 21 July 2021), ConsensusClusterPlus package (version 1.46.0) for consensus clustering analysis, and the MICE command in R for multivariable imputation by chained equation [30].

## 3. Results

There were 158,367 kidney transplant recipients from 2010 to 2019 in the United States. Of these, 3205 (2%) had severely limited functional status with KPS ≤40% at the time of kidney transplant. Therefore, consensus clustering analysis was performed in a total of 3205 functionally impaired kidney transplant recipients. Table 1 shows recipient-, donor-, and transplant-related characteristics of included patients. Most of the included patients (90%) had KPS of 40%. 

Figure 1A shows the CDF plot consensus distributions for each cluster of functionally disabled kidney transplant recipients. The CDF curve showed the best stability for 2 clusters with the curve being flat in the middle part. The delta area plot shows the relative change in the area under the CDF curve (Figure 1B). The largest changes in area occurred between k = 2 and k = 4, at which point the relative increase in area became noticeably smaller. As shown in the CM heat map (Figure 1C, Appendix A), the ML algorithm identified cluster 2 with clear boundaries, indicating good cluster stability over repeated iterations. The mean cluster consensus score was highest in cluster 2 (Figure 2A). In addition, favorable low PACs by both strict and relaxed criteria were demonstrated for 2 clusters (Figure 2B). Thus, using baseline variables at the time of transplant, the consensus clustering analysis identified 2 clusters that best represented the data pattern of our recipients with the KPS of ≤40% at kidney transplant.

### 3.1. Clinical Characteristics of Each Functionally Impaired Kidney Transplant Clusters

There were two distinct clinical clusters identified using consensus clustering analysis. Cluster 1 had 2216 patients (69%), whereas cluster 2 had 989 patients (31%). There were several clinical characteristics between the two clusters, as shown in Table 1 and Figure 3. Kidney transplant recipients in cluster 1 were older in age and more likely to be on dialysis longer prior to transplant and receive a locally allocated standard KDPI deceased donor kidney. In contrast, cluster 2 recipients were younger, had shorter dialysis duration, were more likely to be retransplants, and receive living donors with a lower number of HLA mismatches. Cluster 2 recipients had a higher PRA, less cold ischemia time, and lower proportion of machine-perfused kidneys. 

Overall, very few recipients in clusters 1 and 2 had a working income (8%). Less than half (43% cluster 1, 47% cluster 2) had an undergraduate education or higher. Despite having a low functional status, the majority of recipients received young non-ECD standard KDPI deceased donor kidneys. Only 12% received kidney transplants from ECD donors; high KDPI kidneys were used in 5% of transplants. Thymoglobulin was the most commonly used induction agent (59%).

Appendix A showed the proportion of cluster 1 and cluster 2 based on the OPTN regions. OPTN Regions 7 (12.6%, n = 404), 2 (10.2%, n = 382), 5 (9.3%, n = 299) and 10 (9.2%, n = 296) had the highest number of cluster 1 recipients. Regions 2 (6.0%, n = 193), 2 (7.1%, n = 226), and 10 (3.7%, n = 117) had the highest number of cluster 2 recipients. Regions 7 (19.7%), 5 (13.4%), and 10 (12.9%) had the highest number of recipients from clusters 1 and 2 while regions 6 (0.4%), 1 (2.6%), and 11 (3.8%) had the overall lowest number. 

### 3.2. Posttransplant Outcomes of Each Functionally Disabled Kidney Transplant Cluster

Table 2 shows cluster-based posttransplant outcomes. The 1-year and 5-year death-censored graft survival was 95.9 and 86.9% in cluster 1, and 97.9 and 91.9% in cluster 2 (*p* < 0.001) (Figure 4A). Cluster 1 had lower death-censored graft survival than cluster 2 with HR of 1.92 (95% CI 1.21–3.22) at 1 year and 1.75 (95% CI 1.28–2.40) at 5 years. The 1-year and 5-year patient survival was 93.7 and 79.1% in cluster 1 and 96.5 and 83.9% in cluster 2 (*p* < 0.001) (Figure 4B). Cluster 1 had lower survival than cluster 2 with HR of 1.82 (95% CI 1.26–2.72) at 1 year and 1.45 (95% CI 1.15–1.82) at 5 years. The incidence of 1-year acute allograft rejection was 6.7% in cluster 1, and 3.8% in cluster 2 (*p* = 0.001). Cluster 1 had more acute allograft rejection occurred within 1 year after kidney transplant than cluster 2 with OR of 1.80 (95% CI 1.25–2.60).

## 4. Discussion

Studies have demonstrated poor clinical outcomes after kidney transplant among patients with impaired functional status including reduced patient and allograft survival [1,5,7,8,15,16,17]. As a result, in combination with limited transplant resources, kidney transplant surgeries are uncommonly performed for kidney transplant recipients with severely limited functional status [11]. However, not all kidney transplant recipients with low functional status, including those with KPS ≤40% at the time of transplant, have poor outcomes. Moreover, even for these high-risk recipients, their overall mortality risk remains less when compared to remaining on dialysis [1,11,15]. In this study, unsupervised ML consensus clustering was successfully used to identify two groups of kidney transplant recipients with limited functional status in the U.S. Although small in overall number, these recipients have distinct characteristics and satisfactory post-transplant outcomes and reflect a unique subgroup of kidney transplant recipients. 

The majority (91% in cluster 1 vs. 89% in cluster 2) of patients had KPS of 40% at the time transplant; very few recipients had a KPS <40% (10%, *n* = 308). The overall average recipient age was 51 years. Very few recipients in clusters 1 and 2 had a working income (8%), and less than half (43% cluster 1, 47% cluster 2) had an undergraduate education or higher. Despite having a low functional status, the majority of recipients received young non-ECD standard KDPI deceased donor kidneys. Only 12% received kidney transplants from ECD donors; high KDPI kidneys were used in 5% of transplants. Thymoglobulin was the most commonly used induction agent for both clusters (59%).

Despite the above similarities between the two clusters, there were significant differences in baseline characteristics. Cluster 1 recipients were older compared to cluster 2 (mean 53 years vs. 47 years). Similarly, there was a striking difference in the racial mixture of these recipients; while the number of non-white recipients was low in both groups, cluster 2 had a higher number of white recipients (67 vs. 39%) and significantly lower Black recipients (13 vs. 36%). Compared to cluster 1, cluster 2 recipients had a higher proportion of living donor transplants (48 vs. 13%), less ABDR mismatches (2 vs. 5), and lower cold ischemia times (10 vs. 15 h). The higher number of HLA matches for cluster 2 likely is reflective of living-related kidney donation. 

Compared to cluster 2, cluster 1 recipients had a 1.75-fold increased risk for five-year death censored graft failure and 1.45-fold higher five-year increased risk of death. Inferior outcomes in cluster 1 could potentially be explained by older age and lower rates of preemptive and living donor transplantation. While cluster 2 had higher rate of retransplantation and both clusters received comparable immunosuppression, one-year acute rejection was higher in cluster 1. This finding is possibly a reflection of higher HLA mismatches and longer cold ischemia time in cluster 1 recipients although the observed rates of rejection within both clusters were within the expected standard for kidney transplantation. Higher HLA mismatches are well-known risk factor of acute rejection [34,35,36]. In addition, longer cold ischemia time can lead to increased ischemia reperfusion injury resulting in increased endothelial damage and exposure to donor HLA antigens, and acute rejection [37,38,39,40]. It is also possible that cluster 2 recipients had access to better support and more resources as compared to cluster 1 recipients, given their clinical characteristics of shorter dialysis time and access to living donation. 

To our knowledge, this is the first ML clustering approach successfully applied to kidney transplant recipients low KPS scores in the U.S. Through our use of ML clustering approach, without human intervention or assistance, we were able to identify two distinct clusters of functionally disabled kidney transplant recipients. Cluster 2 recipients do well and have excellent outcomes, which, in itself, is not commonly reported for functionally disabled kidney transplant recipients. These findings from ML clustering approach provide additional understanding towards individualized medicine and opportunities to improve care for vulnerable groups of functionally disabled kidney transplant recipients. Furthermore, there are different cluster distributions among 11 geographic OPTN regions. Regions 7 (19.7%), 5 (13.4%), and 10 (12.9%) had the highest number of recipients from clusters 1 and 2 while regions 6 (0.4%), 1 (2.6%), and 11 (3.8%) had the overall lowest number. When looking at geographic distribution by cluster regions 7 (12.6%), 2 (10.2%), 5 (9.3%), and 10 (9.2%) had the highest number of cluster 1 recipients and regions 2 (6.0%), 7 (7.1%), and 10 (3.7%) had the highest number of cluster 2 recipients. 

Inherent to the source of these data, which came from the UNOS database, there are some limitations to this study. Although individuals were able to be identified as having a low functional status, KPS ≤40%, there is a lack of additional detail regarding these recipients specific to what their limitations were, criteria that centers used to determine their suitability for transplant and information regarding caregiver support. The mean recipient age was relatively low for both clusters 1 and 2 and implies that the nature of the low functional status of these individuals may differ, compared to what is observed for older kidney failure patients with low functional status. Only 39% had diabetes, and 15% had peripheral vascular disease. In addition, we lacked information on interventions and management strategies used by different centers for recipients with low functional status [4,41]. It is also unknown if cluster 2 recipients who received living donors were only offered transplantation through this option. Thus, future studies are required to assess the impacts of interventions and management strategies on changes in functional status and posttransplant outcomes among these two different clusters of functionally disabled kidney transplant recipients. While kidney transplant recipients with lower functional status had an increased mortality after kidney transplant when compared to those with higher functional status [1,11,15], the findings of our study provide further insights into the different allograft and patient outcomes among the unique phenotypic subtypes of kidney transplant recipients with lower functional status, in which those with cluster 1 subtype had the worst posttransplant outcomes in term of death-censored graft failure, death, and allograft rejection. Patients with functionally disabled kidney transplant recipients have different characteristics and should be counseled about their risk of lower posttransplant survival differently. 

## 5. Conclusions

In summary, our ML clustering approach successfully identified two unique phenotypic clusters of kidney transplant recipients with low functional status in the U.S. Each cluster had different characteristics with distinct posttransplant outcomes consisting of allograft rejection, allograft loss, and patient mortality. These findings from ML clustering approach provide additional understanding towards individualized medicine and opportunities to improve transplant opportunities for kidney transplant recipients with low functional status. In addition, our study also showed a varying geographic distribution of these low functional status kidney transplant recipients in the different OPTN Regions in the U.S. Future studies are required to identify strategies to improve outcomes among kidney transplant recipients with lower functional status, especially those with cluster 1 subtype. 

## Figures and Tables

**Figure 1 jpm-12-00859-f001:**
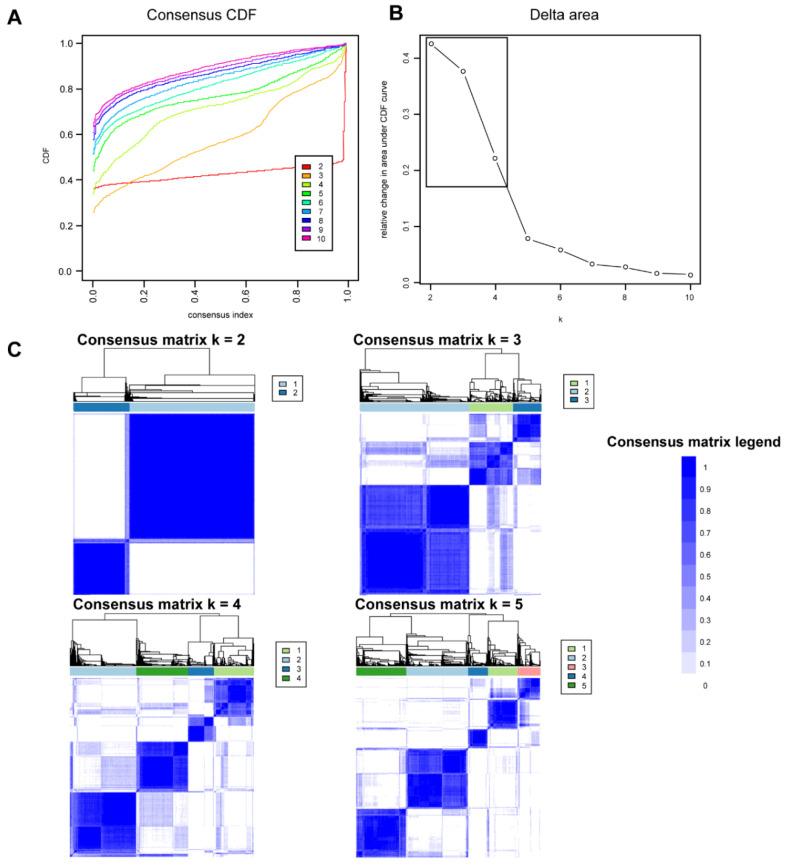
(**A**) CDF plot displaying consensus distributions for each k. (**B**) Delta area plot reflecting the relative changes in the area under the CDF curve. (**C**) Consensus matrix heat map depicting consensus values on a white to blue color scale of each cluster.

**Figure 2 jpm-12-00859-f002:**
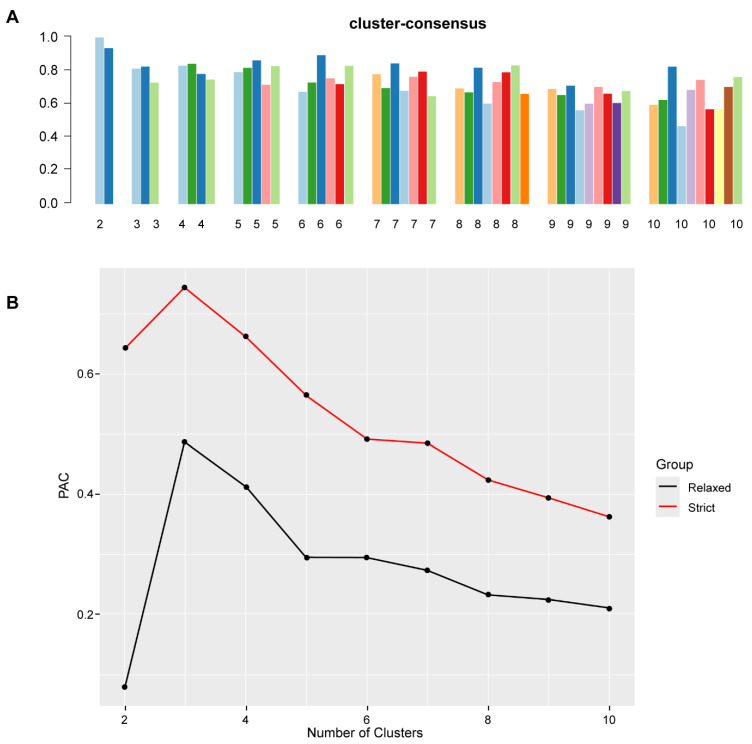
(**A**) The bar plot represents the mean consensus score for different numbers of clusters (K ranges from two to ten). (**B**) The PAC values assess ambiguously clustered pairs.

**Figure 3 jpm-12-00859-f003:**
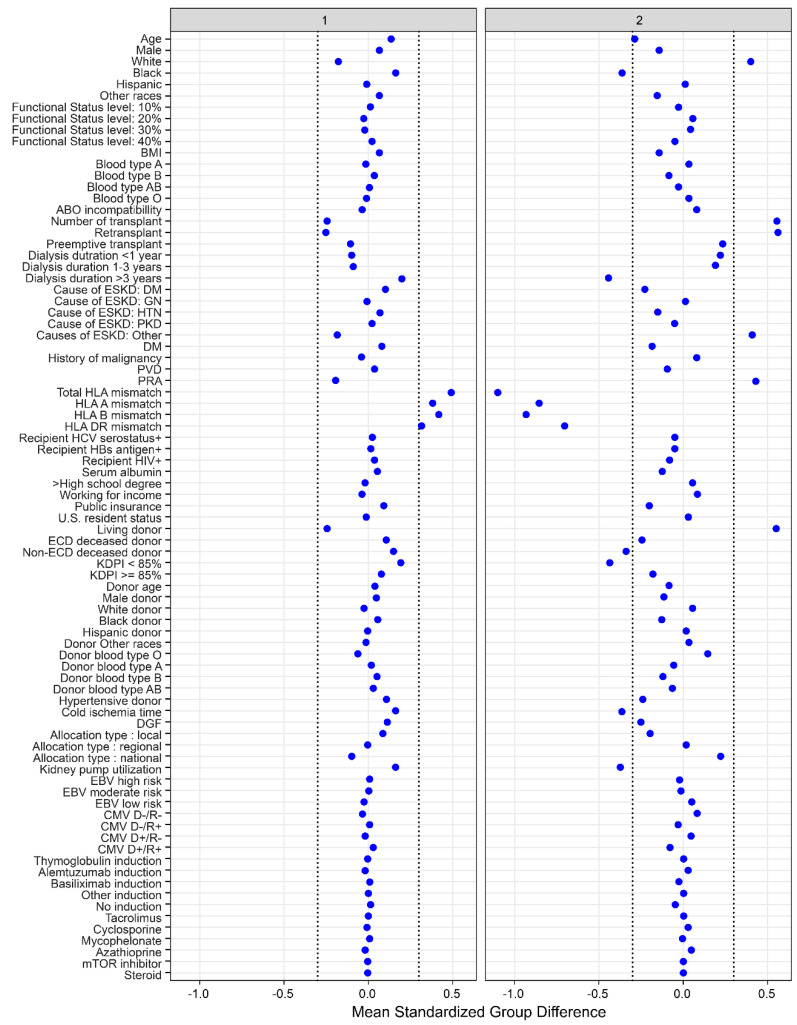
The standardized differences across two clusters for each of baseline parameters. The *x* axis is the standardized differences value, and the *y* axis shows baseline parameters. The dashed vertical lines represent the standardized differences cutoffs of <−0.3 or >0.3. Abbreviations: BMI: Body mass index, CMV: Cytomegalovirus, D: Donor, DGF: Delayed graft function, DM: Diabetes mellitus, EBV: Epstein–Barr virus, ECD: Extended criteria donor, ESKD: End stage kidney disease, GN: Glomerulonephritis, HBs: Hepatitis B surface, HCV: Hepatitis C virus, HIV: Human immunodeficiency virus, HLA: Human leukocyte antigen, HTN: Hypertension, KDPI: Kidney donor profile index, mTOR: Mammalian target of rapamycin, PKD: Polycystic kidney disease, PRA: Panel reactive antibody, PVD: Peripheral vascular disease, R: Recipient.

**Figure 4 jpm-12-00859-f004:**
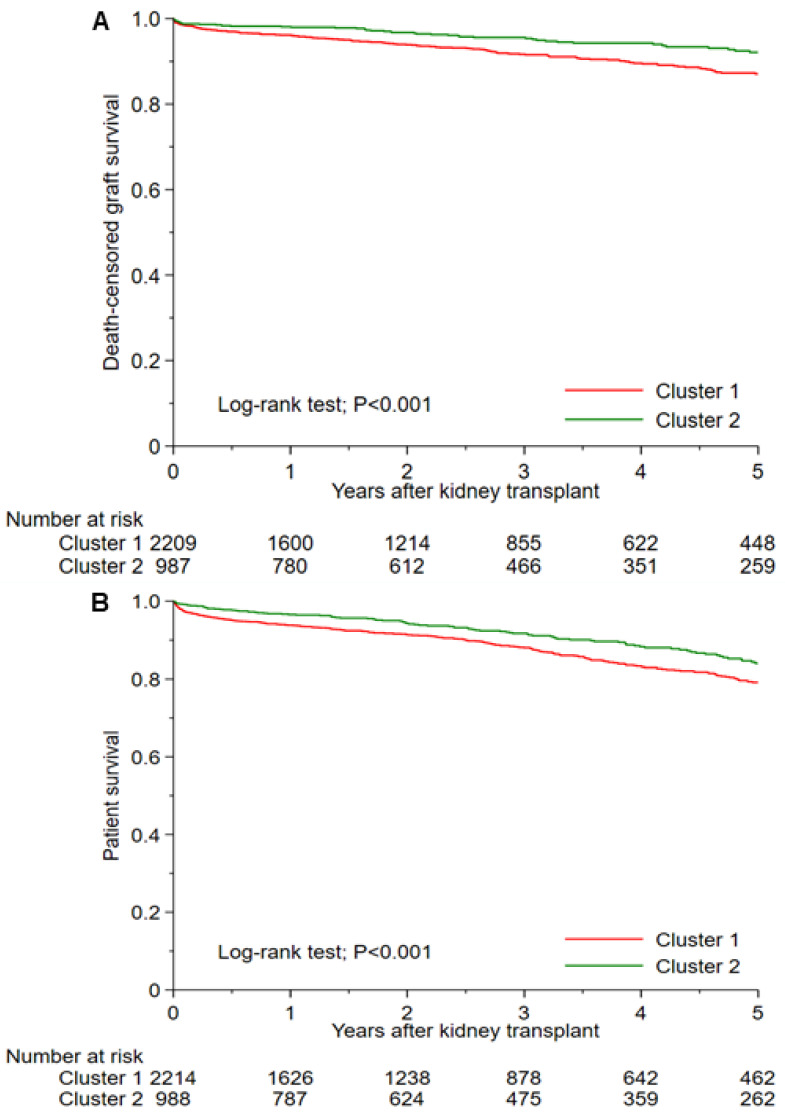
(**A**) Death-censored graft failure and (**B**) patient survival after kidney transplant among two unique clusters of functionally disabled kidney transplant recipients.

**Table 1 jpm-12-00859-t001:** Clinical characteristics according to clusters of functionally disabled kidney transplant recipients.

	All (*n* = 3205)	Cluster 1 (*n* = 2216)	Cluster 2 (*n* = 989)	*p*-Value
Recipient Age (year)	51.0 ± 13.4	52.8 ± 12.7	47.2 ± 14.2	<0.001
Recipient male sex	1957 (61)	1422 (64)	535 (54)	<0.001
Recipient race				<0.001
- White	1521 (48)	854 (39)	667 (67)	
- Black	935 (29)	807 (36)	128 (13)	
- Hispanic	446 (14)	302 (14)	144 (15)	
- Other	303 (9)	253 (11)	50 (5)	
ABO blood group				0.02
- A	1176 (37)	798 (36)	378 (38)	
- B	432 (13)	325 (15)	107 (11)	
- AB	170 (5)	123 (5)	47 (5)	
- O	1427 (45)	970 (44)	457 (46)	
Body mass index (kg/m^2^)	28.4 ± 5.8	28.7 ± 5.7	27.5 ± 5.9	<0.001
Kidney retransplant	392 (12)	89 (4)	303 (31)	<0.001
Dialysis duration				<0.001
- Preemptive	275 (9)	124 (6)	151 (15)	
- <1 year	331 (10)	162 (7)	169 (17)	
- 1–3 years	1860 (58)	1500 (68)	360 (36)	
- >3 years	739 (23)	430 (19)	309 (31)	
Cause of end-stage kidney disease				<0.001
- Diabetes mellitus	1018 (32)	809 (36)	209 (21)	
- Hypertension	643 (20)	506 (23)	137 (14)	
- Glomerular disease	585 (18)	399 (18)	186 (19)	
- PKD	192 (6.0)	145 (7)	47 (5)	
- Other	767 (24)	357 (16)	410 (41)	
Comorbidity				
- Diabetes mellitus	1258 (39)	958 (43)	300 (30)	<0.001
- Malignancy	285 (9)	174 (8)	111 (11)	0.002
- Peripheral vascular disease	485 (15)	368 (17)	117 (12)	<0.001
PRA (%), median (Q25, Q75)	0 (0, 41)	0 (0, 17)	15 (0, 88)	<0.001
Positive HCV serostatus	158 (5)	120 (5)	38 (4)	0.06
Positive HBs antigen	68 (2)	53 (2)	15 (2)	0.11
Positive HIV serostatus	27 (1)	26 (1)	1 (0)	0.002
Functional status				0.04
- 10%	94 (3)	69 (3)	25 (2)	
- 20%	92 (3)	54 (3)	38 (4)	
- 30%	122 (4)	76 (3)	46 (5)	
- 40%	2897 (90)	2017 (91)	880 (89)	
Working income	267 (8)	161 (7)	106 (11)	0.001
Public insurance	2641 (82)	1902 (86)	739 (75)	<0.001
US resident	3192 (99)	2205 (99)	987 (99)	0.23
Undergraduate education or above	1433 (45)	965 (43)	468 (47)	0.04
Serum albumin (g/dL)	3.8 ± 0.6	3.8 ± 0.6	3.7 ± 0.6	<0.001
Kidney donor status				<0.001
- Non-ECD deceased	2067 (64)	1587 (72)	480 (49)	
- ECD deceased	373 (12)	334 (15)	39 (4)	
- Living	765 (24)	295 (13)	470 (47)	
ABO incompatibility	5 (0)	0 (0)	5 (1)	0.003
Donor age (year)	39.8 ± 15.1	40.3 ± 15.5	38.6 ± 14.0	0.004
Donor male sex	1753 (55)	1265 (57)	488 (49)	<0.001
Donor race				<0.001
- White	2289 (71)	1555 (70)	734 (74)	
- Black	419 (13)	330 (15)	89 (9)	
- Hispanic	382 (12)	258 (12)	124 (13)	
- Other	115 (4)	73 (3)	42 (4)	
History of hypertension in donor	710 (22)	588 (27)	122 (12)	<0.001
KDPI				<0.001
- Living donor	765 (24)	295 (13)	470 (48)	
- KDPI < 85%	2267 (71)	1762 (80)	505 (51)	
- KDPI ≥ 85%	173 (5)	159 (7)	14 (1)	
HLA mismatch, median (Q25, Q75)	4 (3, 5)	5 (4, 5)	2 (1, 3)	<0.001
Cold ischemia time (hours)	13.8 ± 9.8	15.3 ± 9.5	10.3 ± 9.5	<0.001
Kidney on pump	1271 (40)	1059 (48)	212 (21)	<0.001
Delay graft function	742 (23)	616 (28)	126 (13)	<0.001
Allocation type				<0.001
- Local	2703 (84)	1937 (87)	766 (77)	
- Regional	226 (7)	150 (7)	76 (8)	
- National	276 (9)	129 (6)	147 (15)	
EBV status				0.08
- Low risk	79 (3)	46 (2)	33 (3)	
- Moderate risk	2782 (87)	1926 (87)	856 (87)	
- High risk	344 (11)	244 (11)	100 (10)	
CMV status				0.001
- D−/R−	540 (17)	342 (15)	198 (20)	
- D−/R+	863 (27)	608 (27)	255 (26)	
- D+/R+	1251 (39)	903 (41)	348 (35)	
- D+/R−	551 (17)	363 (16)	188 (19)	
Induction immunosuppression				
- Thymoglobulin	1893 (59)	1306 (59)	587 (59)	0.82
- Alemtuzumab	346 (11)	230 (10)	116 (12)	0.26
- Basiliximab	631 (20)	446 (20)	185 (19)	0.35
- Other	82 (3)	56 (3)	26 (3)	0.87
- No induction	326 (10)	238 (11)	88 (9)	0.11
Maintenance Immunosuppression				
- Tacrolimus	2967 (93)	2050 (93)	917 (93)	0.83
- Cyclosporine	38 (1)	23 (1)	15 (2)	0.25
- Mycophenolate	2909 (91)	2012 (91)	897 (91)	0.93
- Azathioprine	25 (1)	13 (1)	12 (1)	0.06
- mTOR inhibitors	18 (1)	12 (1)	6 (1)	0.82
- Steroid	1987 (62)	1372 (62)	615 (62)	0.88

Abbreviations: BMI: Body mass index, CMV: Cytomegalovirus, D: Donor, EBV: Epstein–Barr virus, ECD: Extended criteria donor, HBs: Hepatitis B surface, HCV: Hepatitis C virus, HIV: Human immunodeficiency virus, KDPI: Kidney donor profile index, mTOR: Mammalian target of rapamycin, PKD: Polycystic kidney disease, PRA: Panel reactive antibody, R: Recipient. SI conversion: Serum albumin: g/dL × 10 = g/L.

**Table 2 jpm-12-00859-t002:** Post-transplant outcomes according to the clusters of functionally disabled kidney transplant recipients.

	Cluster 1	Cluster 2
1-year death-censored graft failure	4.1%	2.1%
HR for 1-year death-censored graft failure	1.92 (1.21–3.22)	1 (ref)
5-year death-censored graft failure	13.1%	8.1%
HR for 5-year death-censored graft failure	1.75 (1.28–2.40)	1 (ref)
1-year death	6.3%	3.5%
HR for 1-year death	1.82 (1.26–2.72)	1 (ref)
5-year death	20.9%	16.1%
HR for 5-year death	1.45 (1.15–1.82)	1 (ref)
1-year acute rejection	6.7%	3.8%
OR for 1-year acute rejection	1.80 (1.25–2.60)	1 (ref)

## Data Availability

Data are available upon reasonable request to the corresponding author.

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
