# Peer review of "Distinct Phenotypes of Kidney Transplant Recipients in the United States with Limited Functional Status as Identified through Machine Learning Consensus Clustering"

_jpm, 2022, doi:10.3390/jpm12060859_

Round 1

Reviewer 1 Report

Identifying subclasses of a disease could help to individualise the treatment patients need. This paper aims to identify subgroups of kidney transplant patients with limited functional status with different outcomes using the unsupervised consensus clustering for this analysis.

Introduction:

The authors describe the difficulties of selecting those frail patients who are still suitable and would benefit from kidney transplant.

This paper aims to define a subgroup of these patients with low Karnofsky Performance Scale (KPS) who could have good outcome following the transplant. This tool could help the decision making process prior to transplant and also to find the optimal treatment for this high risk group of patients.

The description of the statistical analysis is detailed and the authors provided several references as well.

The authors using the consensus clustering divided the patients into 2 clusters and the outcomes (graft survival, patient survival, risk of rejection) was different in the 2 groups. 

Overall the paper is well written, logical, easy to follow. Tables and plots are clear.

Questions:

1. It is not clear why the unsupervised consensus clustering was chosen for this study. What are the benefits of this method compared to other clustering algorithms such as K-means for example?

2. How can we put these findings into practice? How can I decide which cluster an individual patient would fall into?

3. What is the advantage of this clustering compared to categorising the donors as Standard v Extended Criteria Donor (SCD v ECD) or using the KDPI for risk stratification?

4. Clustering is used to identify subgroups of diseases, (for example it helped to identify a special subgroup of Renal Cell Carcinoma which has different outcome). 

Are these clusters of transplant patients clearly 2 different groups, or what we see is the effect of the different well known risk factors (age, mismatch, diabetes, hypertension etc.).

We already know that kidneys from older donors and older recipients, more mismatches etc have worse outcomes. This analysis confirms these well known facts using a different statistical method. Does this analysis add anything to these?

Author Response

Response to Reviewer#1

Identifying subclasses of a disease could help to individualise the treatment patients need. This paper aims to identify subgroups of kidney transplant patients with limited functional status with different outcomes using the unsupervised consensus clustering for this analysis.

Introduction:

The authors describe the difficulties of selecting those frail patients who are still suitable and would benefit from kidney transplant.

This paper aims to define a subgroup of these patients with low Karnofsky Performance Scale (KPS) who could have good outcome following the transplant. This tool could help the decision making process prior to transplant and also to find the optimal treatment for this high risk group of patients.

The description of the statistical analysis is detailed and the authors provided several references as well.

The authors using the consensus clustering divided the patients into 2 clusters and the outcomes (graft survival, patient survival, risk of rejection) was different in the 2 groups.

Overall the paper is well written, logical, easy to follow. Tables and plots are clear.

Response: Thank you for reviewing our manuscripts and your critical evaluation.

Comment #1

It is not clear why the unsupervised consensus clustering was chosen for this study. What are the benefits of this method compared to other clustering algorithms such as K-means for example?

Response: We appreciate the reviewer’s excellent comment. We have additionally clarified in online supplementary method as the reviewer’s suggestion. Consensus clustering is a more robust approach that relies on multiple iterations of the chosen clustering method on sub-samples of the dataset. By inducing sampling variability with sub-sampling, this provides us with metrics to assess the stability of the clusters and our parameter decisions (i.e., K and linkage) as well as a nice visual component in the form of a heatmap.

We used K-means; Euclidean distance in the Consensus clustering algorithm.  We have additionally included this clarification in the supplementary method as suggested.

Clustering settings used were as follows: maximum number of clusters, 10; number of iterations, 100; subsampling fraction, 0.8; clustering algorithm, K-means; Euclidean distance).

Comment #2

How can we put these findings into practice? How can I decide which cluster an individual patient would fall into?

Response: We appreciate the reviewer’s important comment to improve our manuscript. These findings from ML clustering approach provide additional understanding towards individualized medicine and opportunities to improve care for vulnerable groups of kidney transplant recipients with limited functional status. Regarding clinical practice, we also provided a varying geographic distribution of these low functional status kidney transplant recipients in the different OPTN Regions in the U.S. Future studies are required to identify strategies to improve outcomes among kidney transplant recipients with lower functional status, especially those with cluster 1 subtype. Regarding the individual patient, the characteristics of recipients, donors and transplantation factors are shown in Figure 3. The reviewer made an excellent point and we next step plan to create online model that institutions may use for external validation. The following text has been added in the discussion of the manuscript.   

“In addition, our study also showed a varying geographic distribution of these low functional status kidney transplant recipients in the different OPTN Regions in the U.S. Future studies are required to identify strategies to improve outcomes among kidney transplant recipients with lower functional status, especially those with cluster 1 subtype.” 

Comment #3

What is the advantage of this clustering compared to categorising the donors as Standard v Extended Criteria Donor (SCD v ECD) or using the KDPI for risk stratification?

Response: We appreciate the reviewer’s important comment. We agree that KDPI alone is likely adequate. For comprehensive analysis of unsupervised machine learning (no- multicollinearity concern), we used both Standard v Extended Criteria Donor (SCD v ECD), and KDPI in to models. Kidney transplant recipients in cluster 1 were older in age, more likely to be on dialysis longer prior to transplant and receive a locally allocated standard KDPI deceased donor kidney. In contrast, cluster 2 recipients were younger, had shorter dialysis duration, were more likely to be retransplants, and receive living donors with a lower number of HLA mismatches. Cluster 2 recipients had a higher PRA, less cold ischemia time, and lower proportion of machine-perfused kidneys.

Comment #4

Clustering is used to identify subgroups of diseases, (for example it helped to identify a special subgroup of Renal Cell Carcinoma which has different outcome).

Are these clusters of transplant patients clearly 2 different groups, or what we see is the effect of the different well known risk factors (age, mismatch, diabetes, hypertension etc.).

We already know that kidneys from older donors and older recipients, more mismatches etc have worse outcomes. This analysis confirms these well known facts using a different statistical method. Does this analysis add anything to these?

Response: The reviewer raises important points. The novelty to our work lies in the use of machine learning (ML) into these low functional status kidney transplant recipients in the different OPTN Regions in the U.S. The reviewer is correct that we already know that kidneys from older donors and older recipients, more mismatches have worse outcomes. However, we did not know the clusters of characteristics of donor, recipients, and transplantation of low functional status kidney transplant recipients in the U.S., and this study is the first to bring novelty to demonstrate how characteristics of these patients are, and also provided OPTN Regions map of these two clusters, in order to develop policy to improve outcomes.

Thank you for your time and consideration.  We greatly appreciated the reviewer's and editor's time and comments to improve our manuscript. The manuscript has been improved considerably by the suggested revisions.

Reviewer 2 Report

Very interesting study, well conducted!  The only error I caught was line 317 should state, "functional status of these individuals, may differ or may be different"

Author Response

Response to Reviewer#2

Comment

Very interesting study, well conducted!  The only error I caught was line 317 should state, "functional status of these individuals, may differ or may be different"

Response: Thank you for reviewing our manuscripts and your critical evaluation. The error has been corrected.

Thank you for your time and consideration.  We greatly appreciated the reviewer's and editor's time and comments to improve our manuscript. The manuscript has been improved considerably by the suggested revisions.
